# A Prospective Observational Study on Short and Long-Term Outcomes of COVID-19 Patients with Acute Hypoxic Respiratory Failure Treated with High-Flow Nasal Cannula

**DOI:** 10.3390/jcm12041249

**Published:** 2023-02-04

**Authors:** Kyle J. Medeiros, Carlo Valsecchi, Dario Winterton, Caio A. Morais, Eduardo Diaz Delgado, Shaun Smith, Bijan Safaee Fakhr, Sylvia Ranjeva, Martin Capriles, Timothy Gaulton, Matthew D. Li, Florian Fintelmann, Ismail Tahir, Ryan Carroll, Edward A. Bittner, Kathryn A. Hibbert, Boyd Taylor Thompson, Charles C. Hardin, Roberta RS Santiago, Carolyn J. La Vita, Maurizio Cereda, Lorenzo Berra

**Affiliations:** 1Department of Anesthesia, Critical Care and Pain Medicine, Massachusetts General Hospital, Boston, MA 02114, USA; 2Harvard Medical School, Boston, MA 02115, USA; 3Department of Respiratory Care, Massachusetts General Hospital, Boston, MA 02114, USA; 4Department of Radiology, Massachusetts General Hospital, Boston, MA 02114, USA; 5Department of Pediatric Critical Care Medicine, Massachusetts General Hospital, Boston, MA 02114, USA; 6Department of Medicine, Massachusetts General Hospital, Boston, MA 02114, USA; 7Division of Pulmonary and Critical Care Medicine, Massachusetts General Hospital, Boston, MA 02114, USA

**Keywords:** high-flow nasal cannula, COVID-19, critical care

## Abstract

(1) The use of high-flow nasal cannula (HFNC) combined with frequent respiratory monitoring in patients with acute hypoxic respiratory failure due to COVID-19 has been shown to reduce intubation and mechanical ventilation. (2) This prospective, single-center, observational study included consecutive adult patients with COVID-19 pneumonia treated with a high-flow nasal cannula. Hemodynamic parameters, respiratory rate, inspiratory fraction of oxygen (F_i_O_2_), saturation of oxygen (S_p_O_2_), and the ratio of oxygen saturation to respiratory rate (ROX) were recorded prior to treatment initiation and every 2 h for 24 h. A 6-month follow-up questionnaire was also conducted. (3) Over the study period, 153 of 187 patients were eligible for HFNC. Of these patients, 80% required intubation and 37% of the intubated patients died in hospital. Male sex (OR = 4.65; 95% CI [1.28; 20.6], *p* = 0.03) and higher BMI (OR = 2.63; 95% CI [1.14; 6.76], *p* = 0.03) were associated with an increased risk for new limitations at 6-months after hospital discharge. (4) 20% of patients who received HFNC did not require intubation and were discharged alive from the hospital. Male sex and higher BMI were associated with poor long-term functional outcomes.

## 1. Introduction

Individuals with acute hypoxic respiratory failure (AHRF) often require immediate and escalating respiratory interventions to prevent life-threatening deterioration. High-flow nasal cannula (HFNC), which delivers more than 30 L/min of heated, humidified gas nasally at a fixed oxygen concentration, is a common non-invasive form of respiratory support for individuals whose oxygen requirements exceed the capacity of conventional low-flow systems [1]. Among this population, multiple studies showed improvements in respiratory mechanics, oxygenation, and requirements for intubation and mechanical ventilation with HFNC [2,3]. Since the start of the COVID-19 pandemic, a growing body of literature has supported the use of HFNC among patients with COVID-19-associated AHRF, and international consensus groups have provided updated recommendations supporting the use of HFNC in COVID-19 patients [4,5,6].

However, some clinical studies showed worse outcomes among patients with AHRF that were intubated following failed attempts at non-invasive management [7]. Thus, hesitancy remains surrounding the risk of delaying intubation in patients that may ultimately require mechanical ventilation, and concerns persist regarding the potentially harmful effects of HFNC on lung recovery [8,9].

At present, comprehensive physiological data and long-term follow-up in patients treated with HFNC for AHRF are lacking. In August 2020, a multidisciplinary team at Massachusetts General Hospital implemented a HFNC protocol among patients that failed conventional oxygen supplementation. In this single-center, prospective study, we examined intubation rates and long-term outcomes following HFNC, along with factors associated with long-term functional impairment. We hypothesized that by providing high oxygen concentration and reducing dead space, HFNC provides enough time for antivirals and steroids to be most effective and for the patient to recover from AHRF and avoid intubation.

## 2. Materials and Methods

This prospective observational study evaluated HFNC treatment in patients with COVID-19 pneumonia at the Massachusetts General Hospital. Data collection for this study was approved by the Mass General Brigham Institutional Review Board (protocol #2020P003372). Informed consent was obtained by phone from subjects discharged from the hospital to assess activity of daily living at six months from hospitalization by answering pre-selected questions (Appendix A).

From September 2020 to May 2021, we enrolled all patients older than 18 yearswho were admitted to an intensive care unit (ICU) at our hospital with COVID-19 pneumonia. SARS-CoV-2-related pneumonia was diagnosed based on clinical presentation with hypoxic respiratory failure requiring oxygen supplementation, opacities on chest radiographs, and a positive polymerase chain reaction for the virus obtained via nasopharyngeal or anterior nasal swab. “Failure of the conventional oxygen supply” was defined as a P_a_O_2_/F_i_O_2_ < 200 mmHg, or a S_p_O_2_ lower than 90% with a partial/non-rebreathing mask at flow ≥ 15 L/min (Appendix A). Once started on HFNC, the patient was continuously monitored and reassessed every 2 h. The decision for tracheal intubation was left to the judgement of the treating ICU team, based on the clinical condition of the patient.

Patients were excluded from our analysis if the HFNC treatment lasted less than 12 h and did not require subsequent intubation before hospital discharge, only received HFNC as part of end-of-life care or expressed the wish not to be intubated and resuscitated at hospital admission.

We collected demographic data, time of symptom onset, hospitalization, and ICU admission. Once the decision was made to start HFNC therapy, our team recorded vital signs including hemodynamic parameters, respiratory rate, F_i_O_2_, S_p_O_2_, and the ratio of oxygen saturation to respiratory rate (ROX) score was calculated, prior to HFNC initiation and every 2 h for 24 h. The sequential organ failure assessment (SOFA) score prior to HFNC initiation was computed, the respiratory component of the score was calculated using the SpO_2_/FiO_2_ ratio [10].

The ROX index is defined as the ratio of the oxygen saturation (SpO_2_), to the inspiratory fraction of oxygen (FiO_2_) and the respiratory rate (RR) [11]. Available arterial blood gas data were also collected. For patients that underwent intubation, we recorded intubation time, the respiratory system compliance (C_RS_) and the P_a_O_2_/F_i_O_2_ ratio at the time of intubation and again after 24 h. We collected available laboratory results from the electronic medical record on the start date of HFNC treatment. If more than one result was available, we decided to record the one closest to 8 A.M. More detailed methods are presented in the Appendix A.

Primary endpoints were needed for intubation and mechanical ventilation. Secondary outcomes included hospital and ICU length of stay, length of HFNC treatment and mechanical ventilation, and clinical parameters during HFNC treatment and at the time of intubation. Additionally, we collected mortality data and functional outcomes at 6 months through patient phone calls and access to medical records. If consent was received from the patient, an approved questionnaire (MGB-IRB # 2021P002892) was then administered, either in English or in Spanish, to check the presence of persistent COVID-19 symptoms, use of oxygen supplementation, and the independence in activities of daily living (Appendix A) [12].

Frontal (typically portable) chest radiographs obtained within 48 h before the start of HFNC treatment were analyzed to assess the extent of lung parenchyma involvement. Radiographs were assessed using a previously validated convolutional Siamese neural, network-based approach for an automated assessment of COVID-19 lung disease severity, called the pulmonary x-ray severity (PXS) score (values from 0 to 24, where a higher score indicated more severe lung findings) [13]. Briefly, the deep learning model uses pixel-level image data from frontal chest radiographs as inputs, and outputs a quantitative score for consolidative lung disease severity. This score has been shown to correlate with multiple radiologists’ manual assessment of COVID-19 radiographic disease severity [13,14]. Inference of PXS scores from chest radiographs was performed in the same manner as in the cited literature.

Baseline characteristics, respiratory mechanics and hemodynamic data were compared between the two groups (HFNC success vs. failure), with two-sample parametric or nonparametric tests, as appropriate. The normality of distribution was assessed using the Shapiro-Wilk test. The *t*-test/Wilcoxon rank-sum and chi-square tests were used for comparing groups with continuous/categorical and categorical/categorical variables, respectively. Analyses evaluating the relationship between intubation on outcomes and explanatory variables were performed using logistic regression. A linear mixed model was used to compare the repeated measures during HFNC treatment, with fixed effects on time (hours) and intubation status, and random effect on patients.

Moreover, we used multivariable modeling to identify factors associated with a new limitation in functional capacity (inability to climb 1–2 flights of stairs) at 6 months after hospital admission. We selected six factors a priori due to the small number of expected events, that were based on prior literature and clinical experience [15]. These factors included: demographic factors—age, body mass index, sex (male, female); co-morbidities—diabetes mellitus, asthma; clinical factors—intubation during hospitalization. Age and body mass index were modeled as continuous variables. We developed a multivariable logistic regression model with functional limitation as the dependent variable and the six factors as independent variables. Effect estimates were reported as an odds ratio with 95% confidence intervals (CI).

JMP^®^ Pro 16.0.0 software (Version 16, SAS Institute Inc., Cary, NC, USA, 1989–2021) and R3.6.2 (R Foundation for Statistical Computing, Vienna, Austria) was used for data quality assessment, statistical analysis, and graphics. *p* values < 0.05 were considered statistically significant.

## 3. Results

### 3.1. Short-Term Outcomes with HFNC on Intubation

Over the study period, 187 patients were eligible for HFNC treatment based on the protocol in use at our center. Twenty-six patients were excluded from the analysis since they received HFNC for end-of-life care as a palliative treatment to alleviate air hunger. Eight patients who received HFNC for less than 12 h were not intubated, and were thus excluded. Of the remaining 153 patients, 30 (20%) were successfully managed with HFNC and did not require intubation during their hospital stay. All 30 patients were discharged home alive (100% survival). Intubation occurred in 123 (80%) patients that started HFNC. Failure of HFNC and intubation frequently occurred in the first 48 h (100 out of 123, 81%) (Appendix A). Among the 123 intubated patients, 78 (63%) were discharged alive, while the remaining 45 died (37%) (Figure 1). The overall in-hospital mortality was 29.4% (45 out of 153). One hundred and eight patients were discharged from the hospital alive (30 in the HFNC success group, 78 in the HFNC failure group).

### 3.2. Long-Term Survey

Among the patients discharged home, mortality as an outcome was verified in 100% of the population (Table 1). The 6-month follow-up was completed by 79/108 patients (73%). The remaining 25 patients either did not consent, could not be reached, or had died since hospital discharge. At the 6-month follow-up, four patients had died (three in the HFNC failure group and one in the HFNC success group). The overall mortality at 6 months was 32% (49 out of 153). Without considering the in-hospital mortality, the out-of-hospital mortality alone was less than 4% at 6-months.

One in two respondents returned to work after discharge, although, compared to those who were successfully weaned from HFNC, time to return to work was significantly higher in patients who had undergone endotracheal intubation (7 days vs. 12, respectively, *p* = 0.03) (Table 1). Sixty percent of the patients in the HFNC success group experienced new problems after discharge that they had not experienced before hospitalization, whereas this was found in 81% of the HFNC failure group. The need for supplemental oxygen at the 6-month follow-up was similar between both groups of patients. Hospital readmission was also similar between the two groups. Moreover, almost one in two respondents experienced a new motor or sensory deficit, however, the incidence between the two groups were similar (Table 1).

In our cohort of patients, males had a statistically significant higher probability of experiencing a new limitation in functional capacity; 50% of males experienced this new limitation compared to only 18.2% of females (OR = 4.65, 95% CI [1.28, 20.6]) (Table 2). Notably, each 10 kg/m^2^ increase in BMI increased the probability of experiencing this new limitation over 2.5-fold (OR = 2.63, 95% CI [1.14, 6.76]). Age, history of asthma, diabetes mellitus, and the need for intubation during hospitalization were not statistically significant for experiencing this new limitation (Table 2).

### 3.3. Predictors of HFNC Failure

The time between hospital admission and HFNC initiation was short and did not significantly differ between patients managed successfully with HFNC and those who required intubation (HFNC success 2 (1–2) days vs. failure 1 (0–3) days). The time between symptom onset and hospital admission was also similar between the two groups (HFNC success group 7 (5–10) vs. failure 6 (3–8) days). As shown in Table 3, patients with successful HFNC management had fewer comorbidities, particularly a lower incidence of diabetes mellitus, even though hypertension also tended to be more represented in the failure group (*p*-value 0.06).

Chest radiographs taken within 48 h before the start of HFNC treatment showed no significant difference in PXS scores in both groups. We observed that the ROX index 2 h after initiation of HFNC was higher in the HFNC success group (success: median ROX index at 2 h 4.85 [3.65–6.73] vs. failure: 3.7 [3–4.9], respectively, *p <* 0.01) and showed a significant improvement over time (Figure 2, Table 4).

## 4. Discussion

In this single-center, prospective, observational study, 20% of patients with COVID-19 pneumonia and AHRF, who failed conventional oxygen supplementation and received HFNC according to a protocol in use at our institution, were not intubated during their hospital stay. Additionally, we found that male sex and high BMI were two risk factors associated with an increased risk of experiencing new limitations in daily living 6 months after hospital discharge. The implementation of the HFNC protocol to guide treatment was feasible and was quickly implemented by the ICU team. Before the COVID-19 pandemic, patients who failed conventional oxygen supplementation were subsequently intubated. The HFNC protocol was established to provide an advanced respiratory option to patients who did not necessarily require invasive mechanical ventilation.

HFNC treatment has shown advantages in the treatment of acute respiratory failure. HFNC treatment has been shown to deliver a fixed FiO_2_, enhance patient comfort, decrease work of breathing, and decrease the physiologic dead space [16,17]. These advantages in HFNC treatment could potentially help in avoiding tracheal intubation.

To note, the outcomes of the survivors at the 6-month follow-up were similar between the patients who were successfully managed with HFNC and those who underwent tracheal intubation. In addition, these findings suggest that the 20% of patients who were successfully weaned from HFNC avoided intubation without long-term sequelae. This study is novel as it evaluates the long-term effects of HFNC use on independence in activities of daily living at 6-months post-hospital discharge. No prior studies have evaluated long-term outcomes in patients treated with HFNC for COVID-19 pneumonia.

According to the 6-month follow-up, male sex and higher BMI were the two most significant risk factors that led to long-term sequelae. In our study, males had a higher incidence of experiencing a new limitation post-discharge compared to females. This finding is consistent with prior literature demonstrating a higher incidence and risk of severe disease and death in males with COVID-19 compared to females [18].

Regarding BMI at ICU admission, our data suggests that patients with a higher BMI were more susceptible to experiencing a new limitation at the 6-month follow-up. A previous study published by the U.S. Centers for Disease Control and Prevention found that COVID-19 patients with a BMI greater than 30 had a higher risk of hospitalization, ICU admission, invasive mechanical ventilation, and death [19].

Patients who failed HFNC had significantly higher respiratory rates and, oxygen requirements, and lower ROX index scores at both 12 and 24 h from initiation of HFNC (Table 4). The ROX index is a clinical score that respiratory therapists can easily adopt to assess the respiratory failure of patients at bedside [20]. According to a recent meta-analysis study published by Prakash et al., the ROX index score was shown to be a good indicator of HFNC failure [21]. Among the eight studies and 1301 subjects considered in this meta-analysis, the optimal cutoff for HFNC success vs. failure was close to a ROX index score of 5, within 24 h of HFNC initiation [21]. In our study population, the patients in the HFNC failure group had a median ROX index at 12 h of 4.65 compared to 8.15 in the HFNC success group. The ROX index can help identify patients needing high levels of monitoring and those requiring high levels of respiratory support and mechanical ventilation, especially on admission. Based on our findings, other centers might be able to implement our protocol for the initiation of HFNC with strict monitoring of the patient’s vital signs and the ROX index.

While low respiratory system compliance was associated with increased mortality, levels of oxygenation were similar with the survival group and not associated with increased mortality. Our findings are supported by prior literature in mechanically ventilated patients with acute respiratory distress syndrome. Respiratory compliance and driving pressure reflect the severity of respiratory illness and have been used as a predictor of survival [22]. Levels of hypoxemia, instead, might be determined by multiple respiratory and hemodynamic conditions (e.g., reversible vs. irreversible atelectasis, cardiac output, ventilation/perfusion matching) that are not correlated to survival.

Our study has several limitations. First, this is a single-center, prospective study, which limits the potential generalizability of our findings. Second, this study does not contain a control group and was not randomized. All eligible patients that were put on HFNC at our institution between September 2020 and May 2021 were considered for inclusion. Even though randomization was not utilized, this single-center, prospective study allowed us to validate a HFNC protocol that reduced the number of intubated patients without increasing hospital length of stay (LOS) and long-term neurological conditions, or reducing quality of life, including work- and family-related activities.

## 5. Conclusions

Based on our findings, the need for mechanical ventilation was avoided in 20% of patients treated and successfully weaned from HFNC, and discharged alive from the hospital. Although HFNC was shown to reduce intubation rates among patients with COVID-19 pneumonia admitted to the ICU with AHRF, any delay in intubation can have a deleterious effect on patient outcome. At 6-months post-hospital discharge, male sex and those with a higher BMI were shown to be at increased risk for long-term functional impairment. These findings warrant efforts to optimize treatment and decrease the possibility of bad outcomes in patients at risk.

## Figures and Tables

**Figure 1 jcm-12-01249-f001:**
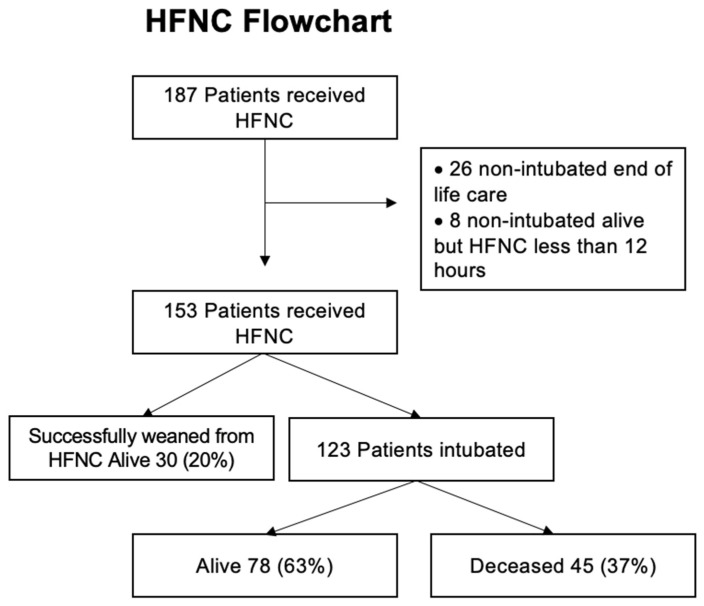
Flow chart. HFNC = High-flow nasal cannula.

**Figure 2 jcm-12-01249-f002:**
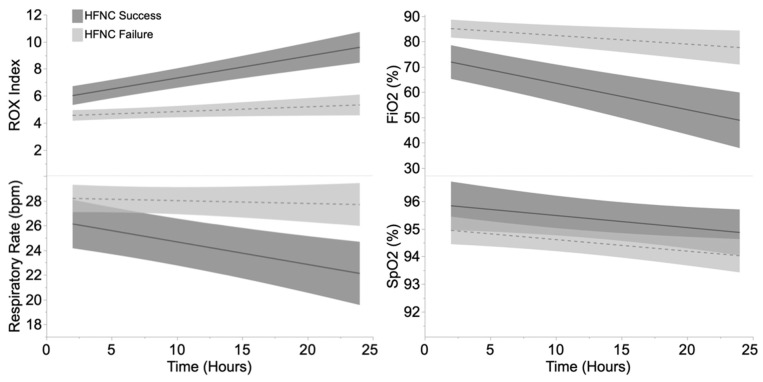
ROX index, respiratory rate (RR), inhaled fraction of oxygen (FiO_2_), and peripheral saturation of oxygen (SpO_2_) during high-flow nasal canula (HFNC) support. The continuous (HFNC success) and dotted (HFNC failure) lines show the values predicted by a linear mixed model.

**Table 1 jcm-12-01249-t001:** Follow-up at 6 months. HFNC = High flow nasal cannula.

	All Subjects	Success (Weaned off HFNC)	Failure (Required Intubation)	*p*
Long term survey				
Discharge from hospital alive, n	108	30	78	
Alive at 6-month follow-up, n (%discharged)	104 (96)	29 (97)	75 (96)	0.90
Age deceased, years				
Median (IQR)	67.5 (57–73)	67	68 (53–75)	
Job before hospital, tot	78	25	53	
Y, n (%)	40 (51)	15 (60)	25 (47)	0.29
Back to work, tot	40	15	25	
Y, n (%)	24 (60)	10 (66)	14 (56)	0.50
Days back, tot	19	7	12	
Median (IQR)		30 (15–60)	75 (38–143)	0.03
New problem after discharge, tot	82	25	57	
Y, n (%)	61 (74)	15 (60)	46 (81)	0.048
Quality of life equal similar, tot	78	25	57	
Y, n (%)	31 (40)	12 (48)	19 (36)	0.31
Able to walk 1–2 flight of stairs, tot	79	25	54	
Y, n (%)	29 (37)	13 (52)	16 (30)	0.06
Any new medications, (tot)	76	24	52	
Y, n (%)	41 (54)	10 (42)	31 (60)	0.14
New medical conditions, tot	80	24	56	
Y, n (%)	26 (33)	7 (29)	19 (34)	0.68
Limitations before, tot	80	26	54	
Y, n (%)	12 (15)	4 (15)	8 (15)	>0.95
New limitations, tot	67	21	46	
Y, n (%)	26 (39)	7 (33)	19 (41)	0.53
Katz, tot	20	6	14	
Median (IQR)		4.5 (4–6)	6 (4–6)	0.37
New oxygen supplementation, tot	81	25	56	
Y, n (%)	13 (16)	3 (12)	10 (18)	0.51
Breathing problem, tot	76	24	52	
Y, n (%)	35 (46)	11 (46)	24 (46)	0.98
Sensory loss, tot	70	24	46	
Y, n (%)	32 (46)	9 (38)	23 (50)	0.71
Motor deficit, tot	74	24	50	
Y, n (%)	36 (49)	11 (46)	25 (50)	0.74
Hospital readmission, tot	87	26	61	
Y, n (%)	29 (33)	8 (31)	21 (34)	0.74
Medical reason, tot	29	8	21	
n (%)	23 (79)	6 (75)	17 (81)	0.72
Reintubated, tot	29	8	21	
n (%)	2 (7)	1 (13)	1 (5)	0.46

**Table 2 jcm-12-01249-t002:** Factors associated with functional capacity limitation at 60 days in 79 survivors who received HFNC for COVID-19 pneumonia.

Covariate	n/Total N (%)	Unadjusted		Adjusted	
		OR (95% CI)	*p*-Value	OR (95% CI)	*p*-Value
Age, years		0.67 (0.48, 0.91)	0.02	0.79 (0.53, 1.15) ^a^	0.24
Sex					
Male	23/46(50.0)	4.50 (1.64, 13.93)	0.005	**4.65 (1.28, 20.6)**	0.03
Female	6/33(18.2)	Reference (1.00)		Reference (1.00)	
Body Mass Index, kg/m^2^		1.5 (0.86, 2.71)	0.16	**2.63 (1.14, 6.76) ^b^**	0.03
Asthma					
Yes	3/18(16.7)	0.27 (0.06, 0.93)	0.057	0.21 (0.03, 1.01)	0.07
No	25/59(42.4)	Reference (1.00)		Reference (1.00)	
Diabetes Mellitus					
Yes	9/35 (25.7)	0.40 (0.15, 1.03)	0.06	0.34 (0.09, 1.10)	0.08
No	20/43 (46.5)	Reference (1.00)		Reference (1.00)	
Intubation during hospitalization					
Yes	16/54(29.6)	0.40 (0.14, 1.03)	0.058	0.39 (0.11, 1.31)	0.13
No	13/25 (52.0)	Reference (1.00)		Reference (1.00)	

a: age estimates in 10-year increments; b: body mass index estimates in 10 kg/m^2^ increments.

**Table 3 jcm-12-01249-t003:** Main demographic information, comorbidities, vital signs, laboratory results and x-ray findings compared between patients weaned off high-flow nasal cannula, and those who experienced HFNC failure and required intubation. Data are presented as mean [±SD], n [%] or median [interquartile range], unless otherwise specified.

	HFNC Success	HFNC Failure	*p*
Demographic information, body max index and HFNC duration			
Subject, n (%)	30 (19.6)	123 (80.4)	
Age, y	62 ± 20	66 ± 13	0.23
Gender (female) n (%	9 (30)	47 (38)	0.4
Race, non-White, n (%)	14 (47)	61 (50)	0.99
Ethnicity non-Hispanic, n (%)	19 (63)	72 (59)	0.76
Body mass index, kg/m^2^	27.5 (23.3–35.1)	29.6 (26.1–34.3)	0.45
HFNC duration, hours	68.5 (28.5–111)	12 (3–28)	<0.01
Comorbidities			
No comorbid disease, n (%)	6 (20)	6 (5)	<0.01
Hypertension, n (%)	19 (63)	94 (80)	0.06
Diabetes mellitus, n (%)	11 (27)	62 (53)	0.01
Chronic kidney disease, n (%)	6 (17)	30 (26)	0.52
Asthma, n (%)	6 (20)	23 (20)	0.98
COPD, n (%)	5 (17)	14 (12)	0.52
Active cancer, n (%)	6 (20)	20 (17)	0.74
HFrEF, n (%)	2 (7)	10 (9)	0.71
Coronary artery disease, n (%)	4 (13)	26 (23)	0.25
Vital signs and x-ray findings			
Oxygen saturation, (%)	93.5 (91–95)	93 (90–96)	0.78
Respiratory rate, breaths/min	26.5 (22.5–30)	32 (26–38)	<0.01
FiO_2_ before HFNC (%)	90 (45–90)	90 (72–90)	0.02
Heart rate, beats/min	85 ±15	89 ±20	0.29
Mean arterial pressure, mmHg	86 (83–101)	89 (82–97)	0.83
SpO_2_/FiO_2_	110 (102–203.5)	106 (100–132)	0.03
Last SpO_2_/FiO_2_ during HFNC	193 (143–238)	99 (95–137)	<0.01
Pulmonary x-ray severity (PXS) score, severe (%)	9 (33)	34 (39)	0.84
Modified SOFA Score	4 (3–6.75)	4 (3–5)	0.359
Laboratory values			
Creatinine	0.83 (0.67–1)	1 (0.82–1.44)	<0.01
Urea	22 (14–26)	21 (16–37)	0.19
White blood cells	8.36 (5.38–11.45)	9.46 (6.27–12.48)	0.23
Platelets	259 (189–325)	223 (167–279)	0.09
Bilirubin	0.4 (0.3–0.6)	0.5 (0.4–0.6)	0.07
C-reactive protein	102.6 (70.7–245.3)	141.5 (73.7–218.6)	0.64

Abbreviations: COPD: Chronic obstructive pulmonary disease; HFNC: High flow nasal cannula; HFrEF: Heart failure with reduced ejection fraction; SOFA: Sequential Organ Failure Assesssment

**Table 4 jcm-12-01249-t004:** Respiratory rate, peripheral saturation of oxygen (SpO_2_), inhaled fraction of oxygen (FiO_2_), and ROX index at 2, 6, 12, and 24 h after the high-f low nasal cannula initiation. Data are presented as median (interquartile range).

Variable	Time (h)	Success (n = 30)	Failure (n = 123)	*p*
Respiratory rate, breath/min	2	25 (22–33)	30 (23–35)	0.21
	6	25 (22–30)	27 (22–31)	0.29
	12	23 (20–25)	26 (23–31)	<0.01
	24	21 (19–28)	24 (20–31)	0.20
SpO_2_, %	2	96 (95–98)	96 (94–97)	0.23
	6	96 (94–98)	94 (92–97)	0.04
	12	95 (94–97)	95 (94–97)	0.77
	24	95 (94–97)	96 (93–97)	0.90
FiO_2_, %	2	69 (60–100)	100 (80–100)	<0.01
	6	60 (50–88)	85 (70–100)	<0.01
	12	55 (40–70)	80 (70–90)	<0.01
	24	50 (40–70)	75 (63–100)	<0.01
ROX Index	2	4.85 (3.65–6.73)	3.7 (3–4.9)	<0.01
	6	6.75 (4–7.88)	4.6 (3.3–5.65)	<0.01
	12	8.15 (5.78–10.65)	4.65 (3.78–5.95)	<0.01
	24	7.2 (6.1–10.38)	5 (3.23–6.95)	<0.01

## Data Availability

Not applicable.

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
