# Peer review of "A Prospective Observational Study on Short and Long-Term Outcomes of COVID-19 Patients with Acute Hypoxic Respiratory Failure Treated with High-Flow Nasal Cannula"

_jcm, 2023, doi:10.3390/jcm12041249_

Round 1

Reviewer 1 Report

I appreciate the authors hard work. however, there are some shortcomings :

1. how the 6 parameters considered for multivariate analysis were elected 

2. in the deceased group, there was a significantly greater incidence of raised creatinine. a cumulative index of comorbidity like Carlson comorbidity index would be better 

3. why were patients reintubated in the follow up. period ?

4. intubation criteria in not mentioned 

5. was prone positioning accorded to all patients on HFNC/Intubated 

Author Response

1.How the 6 parameters considered for multivariate analysis were elected

Thank you for your comment, the 6 parameters considered for the multivariate analysis were selected based of off predictors of long COVID/long terms outcomes based off previous literature (PMID: 33692530).

2. in the deceased group, there was a significantly greater incidence of raised creatinine. a cumulative index of comorbidity like Carlson comorbidity index would be better 

Thank you for your comment, we decided to add a modified SOFA score to the analysis (Table 3). The modified SOFA score was calculated prior to HFNC initiation, the respiratory component of the SOFA score was calculated with the SpO2/FiO2 ratio (PMID: 19242333).

3. why were patients reintubated in the follow up. period ?

Thank you for your comment, two patients were reintubated within the 6 months post hospital discharge. One of the patients was reintubated due to septic shock while the reintubation of the other patient is unknown.

4. intubation criteria in not mentioned 

Thank you for your comment, the need for intubation was left to the discretion of the clinical team based off the clinical condition of the patient. All clinical decisions were left to the clinical care team.

5. was prone positioning accorded to all patients on HFNC/Intubated 

Thank you for this comment, the pronation of a patient was left to the clinical discretion and want of the patient. In a spontaneously breathing patient with a PF ratio below 200, the clinical team will strongly suggestion pronation to the patient, however, it is ultimately up to the patient if they wanted to be prone or not. In a mechanical ventilated patient with a PF ration below 100 pronation was endorsed.

Reviewer 2 Report

The introduction should be reinforced with a greater number of articles referring to the subject. If there is little bibliography on the matter, it can be complemented with the bibliography of the pathology to be treated.

Author Response

The introduction should be reinforced with a greater number of articles referring to the subject. If there is little bibliography on the matter, it can be complemented with the bibliography of the pathology to be treated.

Thank you for this comment, we have added literature to the introduction.

Reviewer 3 Report

The article by Medeiros et al. is of high interest to the medical community in regard to both COVID-19 pandemic, as well as any other form of infection-induced respiratory failure. There are some minor aspects that need to be cleared before the article is considered for publication.

1. Of minor issue. Please recheck the manuscript as there are some formats from the template manuscript still present. Please delete this. 

2. A power analysis is crucial for this type of study and especially in order to validate the results. Please provide this in the methods section. 

3. There was a significant difference in FiO2 before HFNC as demonstrated by the data (see table) between the two groups. This may be a significant bias and thus needs to be discussed in more detail in the discussion section 

Author Response

1.Of minor issue. Please recheck the manuscript as there are some formats from the template manuscript still present. Please delete this

Thank you and we apologize for leaving the template formatting in the submitted manuscript. It has been removed.

2. A power analysis is crucial for this type of study and especially in order to validate the results. Please provide this in the methods section. 

Thank you. We agree with the reviewer that a power analysis is important in studies to ensure that the  sample size is sufficient to detect a difference between groups if such a difference exists. In this case the number of patients in the study was fixed based on availability of COVID patients during a designated interval (wave). Had no difference between groups been detected then we could not conclude whether this finding was due to lack of power or the true absence of a difference between groups. In this study however a clinically meaningful difference was detected making the power analysis unnecessary.

3. There was a significant difference in FiO2 before HFNC as demonstrated by the data (see table) between the two groups. This may be a significant bias and thus needs to be discussed in more detail in the discussion section 

Thank you for this comment, although the difference in baseline FiO2 (before HFNC) was different between the two patients it is represented in the clinical difference between the two groups. The higher the FiO2, the more likely the patient was to fail HFNC.

Reviewer 4 Report

I am very glad that I was asked to review this article.

For first, I would like to note that the conclusions from the study are obvious and there are no novelty from this research (patients with more comorbidities and higher BMI have worse short- and long-term prognosis).

Moreover the conducted studies showed definitely worse ventilation parameters in the group of patients with HFNC therapy failure, what is obvious, but what was the reason for this, other than comorbidities? Because objective parameters, such as the presented PXS, showed no statistical significance between the both groups.

In addition, the presented protocol shows that the decision on intubation was made by the ICU team. But were there used some validated tools to qualify patients, or was it an individual decision, e.g. by the doctor on duty?

There are a lot of minor errors in the text, e.g. the ROX abbreviations have not been expanded and the "How to use this template" section has not been removed.

In my opinion, there are definitely too many authors (22) for the quality of the presented research. For long I haven't seen so many persons in single center study.

Author Response

Moreover the conducted studies showed definitely worse ventilation parameters in the group of patients with HFNC therapy failure, what is obvious, but what was the reason for this, other than comorbidities? Because objective parameters, such as the presented PXS, showed no statistical significance between the both groups.

Thank you. Our study limited was to the design that was chosen. Being an prospective observational study we are unable to explain the causality of the HFNC failure in this group of patients. We can speculate that morbidity, baseline characteristics and respiratory parameters are different in the two groups, however future studies may focus on causality and may unveil the causes that create the difference between the two groups.

In addition, the presented protocol shows that the decision on intubation was made by the ICU team. But were there used some validated tools to qualify patients, or was it an individual decision, e.g. by the doctor on duty?

Thank you for the comment, the decision to intubate the patient was left entirely to the discretion of the clinical care team at the time of HFNC failure.

There are a lot of minor errors in the text, e.g. the ROX abbreviations have not been expanded and the "How to use this template" section has not been removed.

We apologize for the formatting errors. The format template has been removed and other errors, i.e ROX, have been fixed.

In my opinion, there are definitely too many authors (22) for the quality of the presented research. For long I haven't seen so many persons in single center study.

We appreciate this comment and although we agree the number of authors are larger in our study compared to other observational studies, all the authors listed were important and necessary in the completion of this study.

Round 2

Reviewer 4 Report

Thank you for your answer and corrections, but in my opinion the study was not properly designed and therefore the conclusions are too obvious and do not add anything to the current state of knowledge.

Author Response

We appreciate the concern of the reviewer. We highlight below the innovative areas of our manuscript and our intellectual reasoning.   The strength of the methods used in our study is three folds:  - Firstly, the protocol on the use of HFNC was designed by a multidisciplinary team and strictly implemented in the middle of the pandemic in all patients by one to one evaluation of a respiratory therapist with 100% compliance. Patients were closely monitored and the initiation of HFNC was protocolized and started only upon failure of oxygen therapy. Such an approach may be valuable for clinicians implementing similar protocols within their own institutions. - Secondly, the observational, prospectively designed study captured all patients “at risk” within our hospital during the designated study time period and the hypothesis and aims of the study were generated at the time the HFNC protocol was designed.  - Thirdly, patients were followed for 6 months (we are not aware of other studies evaluating long-term effects at 6 months of HFNC).    We acknowledge that our study is not a randomized trial evaluating whether the HFNC is superior to the nasal cannula oxygen. However, our findings represent a great source of information to build future randomized trial on the use of HFNC and provide for the first time some outcomes at 6 months from the incipit of the hypoxemic respiratory failure.  Aware of our study’s limitations, we have been fully transparent and detailed them in our discussion and reported here for you. "Our study has several limitations. First, this is a single-center prospective study which limits the potential generalizability of our findings. Second, this study does not contain a control group and was not randomized. All eligible patients that were put on HFNC at our institution between September 2020 and May 2021 were considered for inclusion. Even though randomization was not utilized, this single center prospective study allowed us to validate a HFNC protocol that reduced the number of intubated patients without increasing hospital length of stay (LOS) and long-term neurological conditions or reducing quality of life, including work and family related activities.”   We hope that the reviewer agrees with us that, despite this is not a definitive trial on the efficacy of HFNC, it provides hypotheses for future studies and it is the first prospective observational study collecting data on 6-months mortality.